# Resveratrol Inhibits Activation of Microglia after Stroke through Triggering Translocation of Smo to Primary Cilia

**DOI:** 10.3390/jpm13020268

**Published:** 2023-01-31

**Authors:** Hongyan Liao, Jiagui Huang, Jie Liu, Yue Chen, Huimin Zhu, Xuemei Li, Jun Wen, Qin Xiang, Qin Yang

**Affiliations:** 1Department of Neurology, the First Affiliated Hospital of Chongqing Medical University, Chongqing 400016, China; 2Department of Neurology, Longevity District People’s Hospital of Chongqing, Chongqing 401220, China

**Keywords:** resveratrol, microglial activation, primary cilia, sonic hedgehog signaling, stroke

## Abstract

Activated microglia act as a double-edged sword for stroke. In the acute phase of stroke, activated microglia might deteriorate neurological function. Therefore, it is of great clinical transforming potential to explore drugs or methods that can inhibit abnormal activation of microglia in the acute phase of stroke to improve neurological function after stroke. Resveratrol has a potential effect of regulating microglial activation and anti-inflammation. However, the molecular mechanism of resveratrol-inhibiting microglial activation has not been fully clarified. Smoothened (Smo) belongs to the Hedgehog (Hh) signaling pathway. Smo activation is the critical step that transmits the Hh signal across the primary cilia to the cytoplasm. Moreover, activated Smo can improve neurological function via regulating oxidative stress, inflammation, apoptosis, neurogenesis, oligodendrogenesis, axonal remodeling, and so on. More studies have indicated that resveratrol can activate Smo. However, it is currently unknown whether resveratrol inhibits microglial activation via Smo. Therefore, in this study, N9 microglia in vitro and mice in vivo were used to investigate whether resveratrol inhibited microglial activation after oxygen-glucose deprivation/reoxygenation (OGD/R) or middle cerebral artery occlusion/reperfusion (MCAO/R) injury and improved functional outcome via triggering translocation of Smo in primary cilia. We definitively found that microglia had primary cilia; resveratrol partially inhibited activation and inflammation of microglia, improved functional outcome after OGD/R and MCAO/R injury, and triggered translocation of Smo to primary cilia. On the contrary, Smo antagonist cyclopamine canceled the above effects of resveratrol. The study suggested that Smo receptor might be a therapeutic target of resveratrol for contributing to inhibit microglial activation in the acute phase of stroke.

## 1. Introduction

Microglia, resident immune cells in the central nervous system, play a local immune surveillance role and are involved in the modulation of the brain’s inflammatory response [1]. After a stroke, microglia are activated, which acts as a double-edged sword. In the acute phase of stroke, activated microglia might deteriorate the outcome through releasing pro-inflammatory factors, such as tumor necrosis factor-α (TNF-α), interleukin-1β (IL-1β), IL-6, and interferon-γ (IFN-γ) [2,3]. In the subacute and chronic phases of stroke, activated microglia promote tissue and vascular remodeling via secreting anti-inflammatory factor IL-10 as well as vascular endothelial growth factor, transforming growth factor-β and brain-derived neurotrophic factor [2,3]. Therefore, it is of great clinical transforming potential to explore drugs or methods that can inhibit abnormal activation of microglia in the acute phase of a stroke to promote the recovery of neurological function after strokes. 

Resveratrol is a naturally occurring polyphenolic phytoalexin extracted from dietary sources such as grapes, mulberries, polygonum cuspidatum, semen cassiae, peanuts, and red wine [4]. More studies have indicated that resveratrol has antioxidant, anti-inflammatory, anti-cancer, anti-apoptotic, anti-aging, and powerful neuroprotective and neurorepair effects [5,6,7,8,9,10,11]. In addition, resveratrol can inhibit the activation of astrocyte and microglia [12,13,14,15,16,17,18]. However, the molecular mechanism of resveratrol-inhibiting microglial activation has not been fully clarified.

Smoothened (Smo) belongs to the Hedgehog (Hh) signaling pathway, which regulates embryonic development, stem cell behavior, axonal and neurite outgrowth, and synaptogenesis in the developing and adult brain [19,20,21]. Smo activation is the critical step that transmits the Hh signal across the primary cilia to the cytoplasm, ultimately resulting in the activation of the glioma-associated oncogene family members (Gli) transcription factors [22,23]. After stroke, infection, trauma, or degeneration of brain, Smo is activated and regulates oxidative stress, inflammation, apoptosis, neurogenesis, oligodendrogenesis, and axonal remodeling, and so on, thus improving neurological recovery [8,9,10,20,24,25,26,27,28,29]. Moreover, more studies have indicated that resveratrol can activate Smo [8,9,10,24,30]. However, it is currently unknown whether resveratrol inhibits microglial activation via activating Smo.

Therefore, in the present study, N9 microglia in vitro and mice in vivo were used to investigate whether resveratrol inhibited microglial activation after oxygen-glucose deprivation/reoxygenation (OGD/R) or middle cerebral artery occlusion/reperfusion (MCAO/R) injury and improved functional outcome via triggering translocation of Smo to primary cilia. The results demonstrated that resveratrol played a neuroprotective role after stroke by partially inhibiting activation and inflammation of microglia, triggering the translocation of Smo to primary cilia and upregulating the protein expression of Shh signaling. On the contrary, cyclopamine, a Smo antagonist, significantly reversed the protective effect of resveratrol.

## 2. Materials and Methods

### 2.1. Culture of N9 Microglia

N9 microglia (kindly provided by the Department of Anesthesia, the Affiliated Children’s Hospital of Chongqing Medical University, Chongqing, China) were cultured in DME/F-12 (Hyclone, Logan, UT, USA) supplemented with 10% FBS (PAN, South America), 100 U/mL penicillin, and 100 U/mL streptomycin at 37 °C in a humidified environment containing 5% CO_2_. When cells reached 80% confluence, they were digested with 0.125% trypsin and subcultured 1:3.

### 2.2. Experimental Animals

Adult male C57 black mice (C57BL/6; 20–25 g) were supplied by the Department of Animal Experiments, Chongqing Medical University. All outcome measurements were performed by observers blinded to the experimental conditions.

### 2.3. MCAO/R Model

Focal cerebral ischemia was induced by transient middle cerebral artery occlusion (MCAO), as described by Estelle Rousselet et al. [31]. Mice were anesthetized with intraperitoneal injection of 1% Pentobarbital Sodium (0.05 mL/10 g; Sigma, St. Louis, MO, USA). The body temperature was maintained at 36.5–37.5 °C with a thermostatically controlled infrared lamp. The surgical area was shaved and prepared with alternating Betadine and ethanol. Following that, the whole process of surgery was performed under the microscope. After a midline incision in the neck, the right common carotid artery (CCA), external carotid artery (ECA), and internal carotid artery (ICA) were exposed and isolated. A small incision was made on the ECA stump, and a heparin-dampened monofilament nylon suture (Beijing Cinontech Co., Ltd., Beijing, China) with a rounded tip was inserted into the ICA through the ECA stump to occlude the origin of right middle cerebral artery (MCA), advanced 8–10 mm, and tightened around the ECA stump by a silk suture. After 1 h of occlusion, the filament was withdrawn. Mice in the sham group were treated with the same surgical procedures except that the filament was not advanced to the MCA origin. Animals were then returned to their cages and closely monitored until they recovered from anesthesia. Mice not exhibiting neurological deficits after reperfusion or who were found with subarachnoid hemorrhage were excluded from this study.

### 2.4. OGD/R Model

The OGD/R model of N9 microglial cells was established according to previously described methods to mimic cerebral artery occlusion and reperfusion injury [32]. Briefly, after three washes with D’Hanks solution, microglia were maintained in D’Hanks solution in a humidified anaerobic incubator (Thermo 3111, Thermo Fisher Scientific Inc., Waltham, MA, USA) in an atmosphere of 94% N_2_, 1% O_2_, and 5% CO_2_ at 37 °C for 150 min. For reoxygenation, D’Hanks solution was replaced with complete medium composed of basal medium, 10% FBS, 100 U/mL penicillin, and 100 U/mL streptomycin, and microglia were incubated in a humidified normoxic atmosphere of 5% CO_2_ at 37 °C for 24 h.

### 2.5. Drugs Treatment

To determine whether effects of resveratrol on activation and inflammation of microglia after stroke via the Smo/Gli-1 signaling, N9 microglia and mice were divided into three or five groups. 

The groups of microglia in vitro were as follows: (1) Normal (Nor) group, microglia were cultured in complete medium without OGD/R. (2) Control (Ctrl) group, microglia were cultured in complete medium for 24 h before OGD/R. (3) Resveratrol (Res) group, microglia were cultured in complete medium containing 20 μmol/L resveratrol (purity 99%, Sigma, St. Louis, MO, USA) for 24 h before OGD/R. (4) Resveratrol combined with cyclopamine (R+C) group, microglia were cultured in complete medium containing 20 μmol/L resveratrol and 5 μmol/L cyclopamine (purity 98%, Cayman Chemical, Ann Arbor, MI, USA) for 24 h before OGD/R. (5) Cyclopamine (Cyc) group, microglia were cultured in complete medium containing 5 μmol/L cyclopamine for 24 h before OGD/R. 

The experimental groups of mice in vivo were as follows: (1) Sham (Sham) group, mice were treated with vehicle (0.5% DMSO) without MCAO/R. (2) Control (Ctrl) group, mice were treated with vehicle (0.5% DMSO) before MCAO/R injury. (3) Resveratrol (Res) group, mice were treated with resveratrol (30 mg/kg) before MCAO/R injury. (4) Resveratrol combined with cyclopamine (R+C) group, mice were treated with resveratrol and cyclopamine (30 mg/kg and 10 mg/kg) respectively before MCAO/R injury. (5) Cyclopamine (Cyc) group, mice were treated with cyclopamine (10 mg/kg) alone before MCAO/R injury. Mice were given the corresponding concentrations of drugs with intraperitoneal injection once a day for 7 days before MCAO/R injury.

### 2.6. Analysis of Neurologic Deficit Scores

Neurologic deficit scores were analyzed at 3 d after MCAO with Longa score [33], modified Bederson score [34], and modified Neurological Severity Score (mNSS) [35] by an independent investigator in a blinded fashion. Longa score was used to determine motor motion functions which were graded on a scale of 0–4 (0, no deficits; 4, no spontaneous walking and decreased level of consciousness). A modified Bederson score was used to evaluate global neurological functions which was graded on a scale of 0–5 (0, no deficits; 5, no movement). The Modified Neurological Severity Score (mNSS) is an amalgamation of motor, sensory, reflex, and balance tests. It was graded on a scale of 0–18 (normal scoyure 0; maximal deficit score 18). The lower the score, the less severe the damage.

### 2.7. Determination of Cerebral Infarct Volume

Infarct volume was measured at 24 h after MCAO according to previously described methods [10]. Brains (*n* = 4 for each group) were rapidly dissected and frozen for 20 min at −20 °C. Then, the brains were coronally sectioned into five consecutive slices of 2 mm thickness and incubated in a 2% solution of 2,3,5-triphenyltetrazolium chloride (TTC; Solarbio LIFE SCIENCE, Beijing, China) at 37 °C in the dark for 30 min, followed by immersion in 10% paraformaldehyde. TTC-stained sections were photographed. The lesion volumes were calculated by the following equation: %HLV = {[total infarct volume − (right hemisphere volume − left hemisphere volume)]/left hemisphere volume} × 100%. 

### 2.8. Enzyme-Linked Immunosorbent Assay (ELISA)

The levels of TNF-α, IL-1β, and IL-10 in cell supernatants and in brain homogenate were measured by using enzyme-linked immunosorbent assay kits (NEOBIOSCIENCE, China) following the manufacturer’s instructions. 

### 2.9. Immunocytochemistry

Microglia were seeded in coverslips with poly-L-lysine pretreatment in each group. Cells were fixed with 4% formaldehyde solution for 30 min at room temperature, incubated with 1% Triton X-100 for 30 min, and blocked with 5% goat or donkey serum for 1 h at 37 °C. Subsequently, cells were incubated overnight at 4 °C with the following primary antibodies: monoclonal rabbit anti-ionized calcium binding adaptor molecule1(anti-Iba1) antibody (1:100, Abcam, Cambridge, UK), monoclonal mouse anti-acetylated tubulin (anti-Ac-Tu) antibody (1:200, Sigma, St. Louis, MO, USA), polyclonal rabbit anti-Gli-1 antibody (1:100, Abcam, UK), and rabbit anti-Smo antibody (1:200, Abcam, UK). After washing three times with PBS, cells were reacted with the following fluorescent secondary antibodies at 37 °C for 1 h: Cy3-conjugated goat anti-rabbit IgG (1:200, Proteintech, Rosemont, IL, USA), FITC-conjugated goat anti-mouse IgG (1:200, Proteintech, Rosemont, IL, USA). The primary antibodies were replaced with PBS in the negative controls. Nuclei were counterstained with 4,6-diamidino-2-phenylindole (DAPI, Beyotime, China) for 5 min in the dark. Finally, all images were observed and acquired using an A1+R laser confocal microscope (Nikon, Tokyo, Japan) or TH4-200 fluorescence microscopy (Olympus, Tokyo, Japan).

### 2.10. Western Blotting

To analyze cellular protein levels, the cultured cells were digested with 0.25% trypsin and then centrifuged at 1500 rpm for 15 min. The samples were homogenized in radioimmunoprecipitation assay (RIPA) lysis buffer (Beyotime, Shanghai, China) containing 1% phenylmethane sulfonyl fluoride (PMSF; Beyotime, Shanghai, China), and incubated on ice for 30 min. After centrifugation (12,000 rpm and 4 °C) for 10 min, the protein concentration of each extract was measured using a BCA Protein Assay Reagent Kit (Beyotime, Shanghai, China). A total of 40 μg protein per lane was loaded and separated via SDS-PAGE, and then transferred onto polyvinylidene difluoride membranes (EMD, Millipore, Billerica, MA, USA). The membranes were blocked with 5% non-fat milk at room temperature for 2 h and then incubated overnight at 4 °C with the following primary antibodies: monoclonal rabbit anti-Iba1 (1:1000, Abcam, UK) antibodies; polyclonal rabbit anti-Shh, -Ptch-1, -Smo, -Gli-1 (1:1000, Abcam, UK) antibodies; rabbit anti-GAPDH (1:1000; Beyotime, China) antibody was a loading control. The membrane was washed with TBST buffer for three times and incubated with horseradish peroxidase-conjugated affiniPure goat anti-rabbit or anti-mouse IgG (1:2000, Beijing Zhongshan Golden Bridge, China) at 37 °C for 1 h. After washing with TBST buffer three times, protein bands were detected by enhanced chemiluminescence and their intensities were analyzed with quantity One software (Bio-Rad, Hercules, CA, USA).

### 2.11. Statistical Analysis

All the values were expressed as the mean ± standard deviation and statistically analyzed with SPSS 22.0. Statistically significant differences of biochemical data in different groups were evaluated by One-way ANOVA analysis of variance followed by LSD post-hoc test for multiple comparisons. The difference of *p* < 0.05 was considered to be statistically significant. For each assessment, at least three independent experiments were performed.

## 3. Results

### 3.1. Resveratrol Decreased Infarct Volume, Improved Neurological Function, and Inhibited Activation of Microglia and Inflammation after Stroke

Infarct volume with TTC staining was examined at 24 h after MCAO/R injury. Neurological functional outcome with Longa score, Bederson score, and mNSS were examined at 3 d after MCAO/R injury in mice. As shown in Figure 1, in the sham group, there were no cerebral infarct and neurological deficits. In the control group, all mice exhibited significantly extensive lesion in the striatum and lateral cortex and neurological deficits with Longa score, Bederson score, and mNSS (*p* < 0.05; Figure 1A–E). However, the infarct volume, Longa score, Bederson score, and mNSS in the resveratrol group after stroke were decreased significantly than those in the control group (*p* < 0.05; Figure 1A–E). 

Microglial activation plays a pivotal role in damage and repair of ischemic stroke [3]. Morphological change is one of the classic features of microglia activation and Iba1 is a marker of activated microglia [2]. Immunofluorescence after MCAO/R injury in vivo showed that microglia in the sham group had small cell bodies, thin-branched processes, and quite weak fluorescence. While in the control group, microglia had swollen bodies, thick-branched processes, and stronger immunofluorescence intensity (Figure 2A). In the resveratrol group, microglial bodies, branched processes, and immunofluorescence intensity of microglia were significantly smaller, thinner, and weaker than those in the control group. Besides, the expression of Iba1 protein by western blotting was significantly increased in the control and resveratrol groups than those in the sham or normal group after MCAO/R or OGD/R injury, and it was significantly decreased in the resveratrol group than those in the control group (*p* < 0.05; Figure 2B,C,G,H). 

Moreover, neuroinflammation is closely related to the release of inflammatory factors by activated microglia, especially the pro-inflammatory cytokines TNF-α and IL-1β and the anti-inflammatory cytokine IL-10 [3]. ELISA analysis showed that protein levels of TNF-α, IL-1β, and IL-10 at 3d after MCAO/R injury in vivo or at 24 h after OGD/R injury in vitro were significantly increased in the control and resveratrol groups than those in the normal or sham group (*p* < 0.05; Figure 2D–F,I–K). However, TNF-α and IL-1β were significantly decreased, and IL-10 was significantly increased in the resveratrol group compared to those in the control group (*p* < 0.05; Figure 2D–F,I–K). 

### 3.2. Resveratrol Triggered the Translocation of Smo/Gli-1 Molecules and Upregulated the Expression of Shh, Ptc-1, Smo, and Gli-1 after OGD/R Injury of N9 Microglia In Vitro

Shh regulates a variety of processes such as embryogenesis, cell proliferation and differentiation, and tissue repair during inflammation [21,22]. The components of the Shh signaling pathway in mammals include Shh, Ptc-1, Smo, and Gli-1. Moreover, primary cilium is a necessary “organ” for the canonical Shh signaling transduction in vertebrates [24]. Therefore, the present study examined whether N9 microglia possessed the primary cilia harboring Shh signaling molecules and resveratrol could trigger the translocation of Smo/Gli-1 molecules and upregulate the expression of Shh, Ptc-1, Smo, and Gli-1.

Acetylated a-tubulin (Ac-tu) is a marker of the primary cilia. Immunofluorescence assay showed that microglia had a primary cilium. In the normal and control groups, Smo lay in cytoplasm. In the resveratrol group, Smo translocated into the primary cilia (Figure 3A). In addition, Gli-1, which accumulated in cytoplasm in the normal group, transferred to the nuclei from the cytoplasm in the control and resveratrol groups (Figure 3B). Moreover, Western blotting showed that the expression of Shh, Ptc-1, Smo, and Gli-1 proteins significantly increased in the resveratrol group compared tp those in the control group (*p* < 0.05, Figure 3C–G). 

### 3.3. Cyclopamine Increased Infarct Volume, Deteriorated Neurological Function, and Promoted Activation of Microglia and Inflammation after Stroke

Next, we replaced resveratrol with cyclopamine, a Smo receptor inhibitor, which inhibited the Shh signaling system. We found that infarct volume, Longa score, Bederson score, and mNSS were significantly increased compared to those in the resveratrol group after MCAO/R injury in vivo (*p* < 0.05; Figure 4A–E). Moreover, compared with the resveratrol group, the Iba1, TNF-α, and IL-1β proteins were significantly increased, while the IL-10 was decreased in the cyclopamine group in vitro or in vivo (*p* < 0.05; Figure 4G–P). Furthermore, Smo lay in cytoplasm, Gli-1 accumulated in cytoplasm, and the expression of Shh, Ptc-1, Smo, and Gli-1 proteins significantly decreased in the cyclopamine group compared to those in the resveratrol group (Figure 5A–G). 

## 4. Discussion

The present study demonstrated that resveratrol could inhibit activation of microglia and inflammation, decrease infarct volume, and improve neurological functional outcome in the acute phase of stroke. Moreover, N9 microglia possessed primary cilia harboring Shh signaling molecules and resveratrol triggered translocation of Smo/Gli-1 molecules and upregulated the expression of Shh, Ptc-1, Smo, and Gli-1 after OGD/R injury of microglia in vitro. On the contrary, Smo antagonist cyclopamine canceled the above effects of resveratrol. The study suggested that Smo receptor might be a therapeutic target of resveratrol for inhibiting microglial activation in the acute phase of stroke. 

Many literatures have reported that resveratrol has beneficial and harmful effects based on the characteristics of the enrolled patients, the doses used, duration of resveratrol supplementation, the environment stayed, and so on [6,36]. Wang et al. reported that 0.1, 1, and 2.5 μmol/L resveratrol promoted viability and proliferation of human umbilical cord derived MSCs (hUC-MSCs), while 5 and 10 μmol/L resveratrol reversed the above effects [37]. Similar finding reported that resveratrol with low concentration (50 µM) and short time (24 h) treatment had no significant effect on cell viability in normal cells (the renal tubular epithelial cell line HK-2). After treating for longer, a significant inhibitory effect appeared, while for cancer cells (murine mammary carcinoma cell line 4T1), resveratrol could significantly decrease cell viability in a dose-dependent manner all the time [38]. At the same time, others found that under low pH conditions, resveratrol promoted growth inhibition, internucleosomal DNA fragmentation, and apoptosis on human pancreatic cancer cell lines, but not on normal epithelial cells [39]. In addition, recent literature reported that resveratrol inhibited proliferation, promoted apoptosis, and induced autophagy by targeting HIF-1α or blocking SREBP1 expression [40,41]. However, our previous research demonstrated that resveratrol promoted the proliferation of neural stem cells, reduced apoptosis, and improved neurological function after stroke in vivo or in vitro [8,10,32]. Moreover, the present study verified that resveratrol could inhibit activation of microglia and inflammation after stroke to exert neuroprotective effect. Therefore, these studies suggest that extensive physiological, pharmacological or toxicological effects of resveratrol are related to the concentration of resveratrol, the physiological, pathological state, and types of organs, tissues, or cells. It needs to be extensively investigated.

It is reported that the neuroprotective mechanism of resveratrol is associated with various pathways which medicate the anti-oxidative, anti-apoptotic, and anti-inflammatory properties [6,42]. Recent literatures reported that resveratrol could inhibit oxidation and apoptosis in neurodegenerative diseases through the Nrf2/HO-1, SIRT1, RAX/P-PKR, and JAK2/STAT3/PI3K/AKT/mTOR signaling pathways [43,44,45,46]. Besides, a growing body of in vitro and in vivo evidence indicates that resveratrol acts anti-inflammatory property through multiple pathways. For example, resveratrol inhibited microglial activation and neuroinflammation, and reduced morphine tolerance and trigeminal neuralgia via AMPK signaling [12,13]. In the LPS/IFN-γ-treated N9 microglia, resveratrol attenuated microglial activation via SIRT1-SOCS1 pathway [14]. In experimental subarachnoid hemorrhage, resveratrol decreased early brain injury, neuroinflammation and microglia activation through inhibition of NLRP3 inflammasome activation or TLR4 pathway [15,16]. Zhao and Qi et al. reported that resveratrol could alleviate the ethanol and Aβ1-42-induced neuroinflammation and microglial activation through inhibition of TLR2-MyD88-NF-κB and NF-κB signal pathways [17,18]. In addition, our present study indicated that resveratrol could inhibit activation of microglia and neuroinflammation via triggering the translocation of Smo and Gli-1 molecules and upregulating the expression levels of Shh, Ptc-1, Smo, and Gli-1 in the acute phase of stroke. Therefore, these studies all suggest that resveratrol can suppress the oxidation, apoptosis, and inflammation via triggering a cascade of neuroprotective pathways. 

Primary cilium is a non-motile microtubule-based, centriole-derived, membrane-ensheathed process present in most mammalian cells that regulates all kinds of physiological functions, including cell division and metabolism [47]. Defects in primary cilia lead to a variety of diseases in humans, such as cancers and obesity. In the central nervous system, it is well documented that neurons, astrocytes, neural stem cells, and some choroid plexus cells contain a single primary cilium [48,49,50]. Do microglia have primary cilia? Sipos et al. reported that microglia in mouse hippocampus do not display primary cilia marked by type 3 adenylyl cyclase (AC3) and ADP-ribosylation factor-like 13B (ARL13B) [51]. However, our study, for the first time, showed that N9 microglia had a primary cilium marked by acetylated a-tubulin (Ac-tu). Moreover, resveratrol triggered Smo to translocate into primary cilium from cytoplasm and Gli-1 to translocate into nuclei from cytoplasm and upregulate the expression of Shh, Ptc-1, Smo, and Gli-1. In the future, we will test whether microglia in development and adult normal and pathological brain have primary cilia, and elucidate their functions and molecular pathways in the mature brain.

Smo is one of the critical components of Hh signaling pathway. Primary cilia are required for Smo to transduce vertebrate Shh signals [24]. The Shh ligand relieves Patched1 repression of Smo, allowing Smo to accumulate in cilia and activate Gli transcription factors [22], but how Smo accumulates in cilia and is activated is incompletely understood. Raleigh et al. reported that cilia-associated oxysterols bind to two distinct Smo domains to modulate Smo accumulation in primary cilia and activate Hh pathway, and the oxysterol synthase HSD11β2 participates in the production of Smoothened-activating oxysterols and promotes Hh pathway activity [52]. Deshpande et al. reported that membrane sterols stimulate Smo to drive Hh pathway activity [53]. Our previous and present studies in bone mesenchymal stem cells, NIH3T3 cells, and microglia showed that resveratrol can trigger the translocation of Smo to primary cilia from cytoplasm and activate the Shh pathway [24,54]. Therefore, it is critical to elucidate how Smo accumulates in primary cilia.

Taken together, the present study showed that N9 microglia possessed primary cilia harboring Shh signaling molecules. Resveratrol could inhibit activation of microglia and inflammation, decrease infarct volume, and improve the neurological functional outcome in the acute phase of stroke, trigger the translocation of Smo to primary cilia from cytoplasm, and upregulate the expression levels of Shh, Ptc-1, Smo, and Gli-1 after OGD/R injury of microglia in vitro. On the contrary, Smo antagonist cyclopamine canceled the above effects of resveratrol. The findings of the present study are important for understanding the mechanism underlying the resveratrol-inhibited activation of microglia and the improved neurological functional outcome in the acute phase of stroke. In the future, investigations are required to determine whether microglia in development and adult normal and pathological brain have primary cilia, and how resveratrol triggers the translocation of Smo to primary cilia.

## Figures and Tables

**Figure 1 jpm-13-00268-f001:**
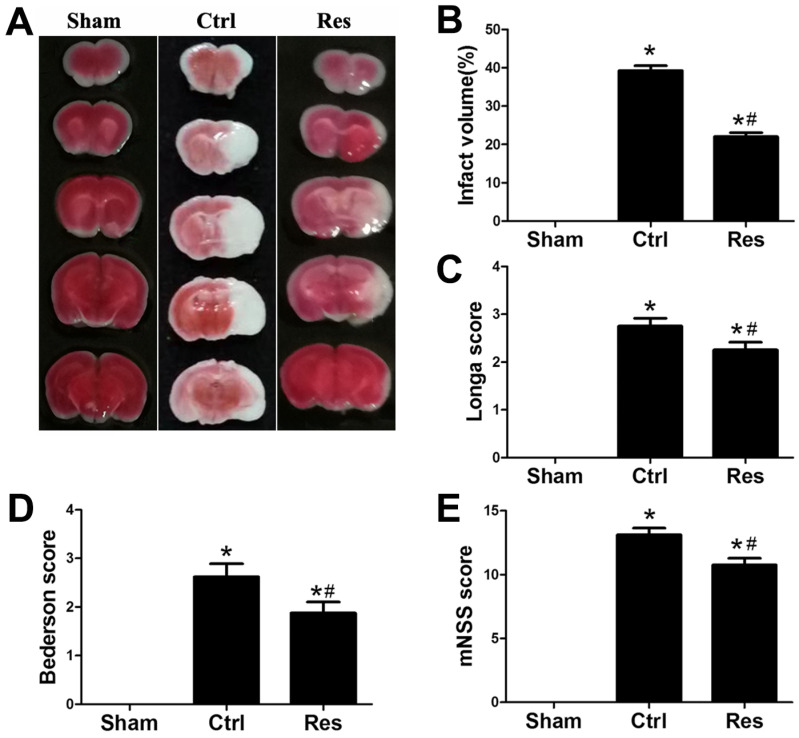
Effects of resveratrol on infarct volume and neurological functional score after MCAO/R injury in vivo. (**A**,**B**) Quantification of infarction volumes with TTC staining at 24 h after MCAO/R injury in vivo. (**C**–**E**) Neurological function score with Longa score (**C**), Bederson score (**D**), and mNSS (**E**) were assayed at 3 d after MCAO/R injury in vivo. * *p* < 0.05 vs. Nor; # *p* < 0.05 vs. Ctrl; (ANOVA, *n* = 6 each group).

**Figure 2 jpm-13-00268-f002:**
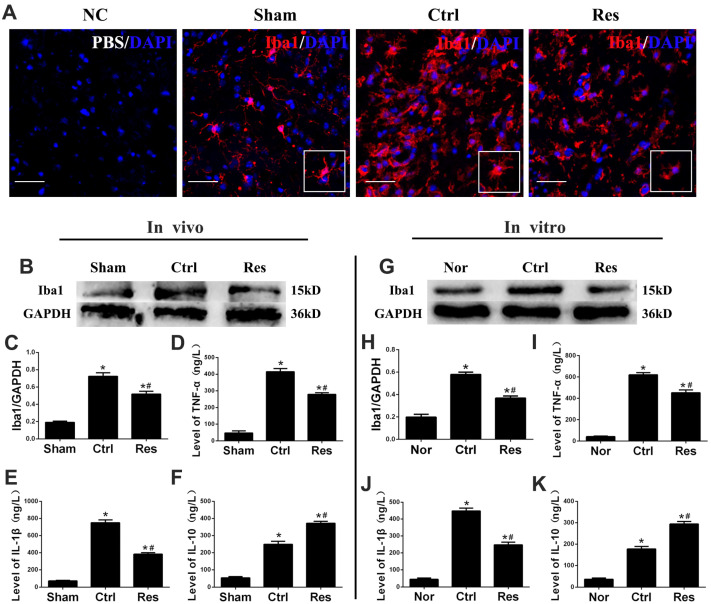
Resveratrol inhibited activation of microglia and inflammation after MCAO/R injury in vivo or OGD/R injury in vitro. (**A**) Analysis of immunofluorescent images showed activation of microglia in each group after MCAO/R injury. NC: Negative control, Iba1 antibody was replaced with PBS to serve as a negative control. Iba1, red; DAPI, blue. Scale bars = 50 μm. (**B**,**C**,**G**,**H**) Resveratrol downregulated expression of Iba1 protein in vivo or in vitro with western blotting assay. (**D**–**F**,**I**–**K**) ELISA was used to measure the levels of TNF-α, IL-1β, and IL-10 in brain homogenate and in the cell supernatant of microglia. * *p* < 0.05 vs. Nor; # *p* < 0.05 vs. Ctrl. (ANOVA, **B**–**F**: *n* = 6 each group; **G**–**K**: *n* = 3 each group).

**Figure 3 jpm-13-00268-f003:**
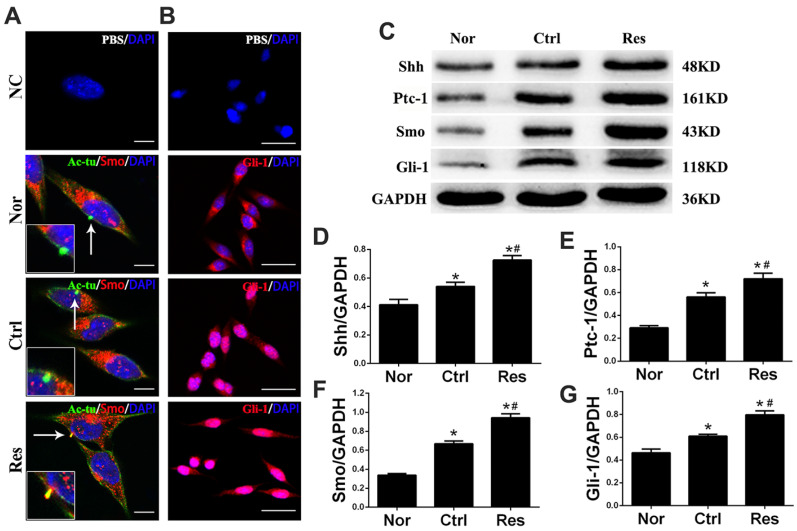
Resveratrol triggered the translocation of Smo/Gli-1 molecules and the Shh signaling in microglia after OGD/R injury in vitro. (**A**) Resveratrol promoted Smo to translocate into primary cilia from cytoplasm in microglia after OGD/R injury in vitro. In the Nor or Ctrl group, Smo lay in cytoplasm. In the Res group, Smo translocated into the primary cilia. NC: Ac-tu and Smo antibodies were replaced with PBS. Primary cilia were labeled with Ac-tu, green; Smo, red; DAPI, blue. Scale bars = 10 μm. (**B**) Resveratrol promoted nuclear translocation of Gli-1 in microglia after OGD/R injury. In the Nor group, Gli-1 accumulated in cytoplasm. In the Ctrl group, Gli-1 partly transferred to the nucleus. In the Res group, Gli-1 almost transferred to the nucleus. NC: Gli-1 antibody was replaced with PBS to serve as a negative control. Gli-1, red; DAPI, blue. Scale bars = 50 μm. (**C**–**G**) Resveratrol-upregulated Shh, Ptc-1, Smo, and Gli-1 expression in microglia after OGD/R injury with Western blotting assay. * *p* < 0.05 vs. Nor; # *p* < 0.05 vs. Ctrl. (ANOVA, *n* = 3 each group).

**Figure 4 jpm-13-00268-f004:**
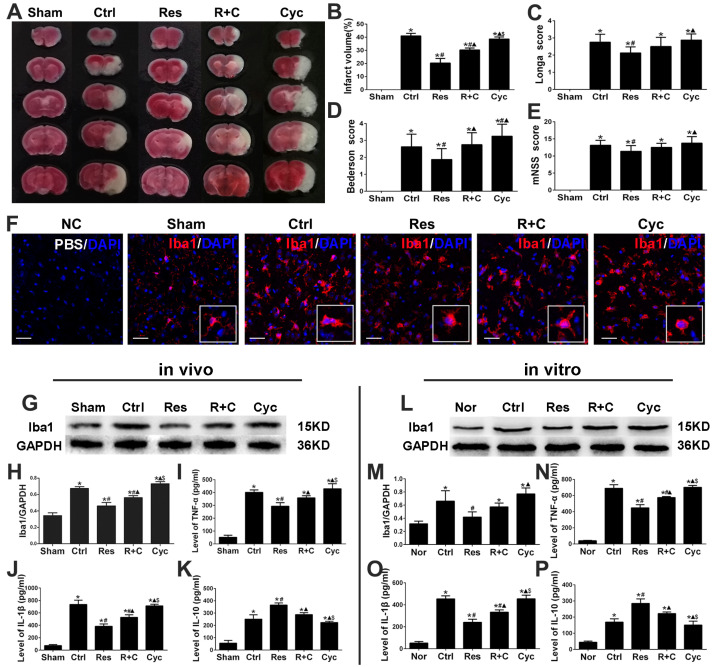
Cyclopamine deteriorated infarct volume, neurological functional score, microglia activation, and inflammation after stroke. (**A**,**B**) Quantification of infarction volumes with TTC staining at 24 h after MCAO/R injury in vivo. (**C**–**E**) Cyclopamine aggravated Longa score, Bederson score, and mNSS at 3 d after MCAO/R injury. (**F**) Analysis of immunofluorescent images showed activation of microglia in each group after MCAO/R injury. Scale bars = 50 μm. (**G**–**P**) The expression of Iba1 protein and Inflammatory factors in vivo or in vitro were measured by western blotting and ELISA. * *p* < 0.05 vs. Nor; # *p* < 0.05 vs. Ctrl; ▲ *p* < 0.05 vs. Res; $ *p* < 0.05 vs. R+C (ANOVA, **A**–**K**: *n* = 6 each group; **L**–**P**: *n* = 3 each group).

**Figure 5 jpm-13-00268-f005:**
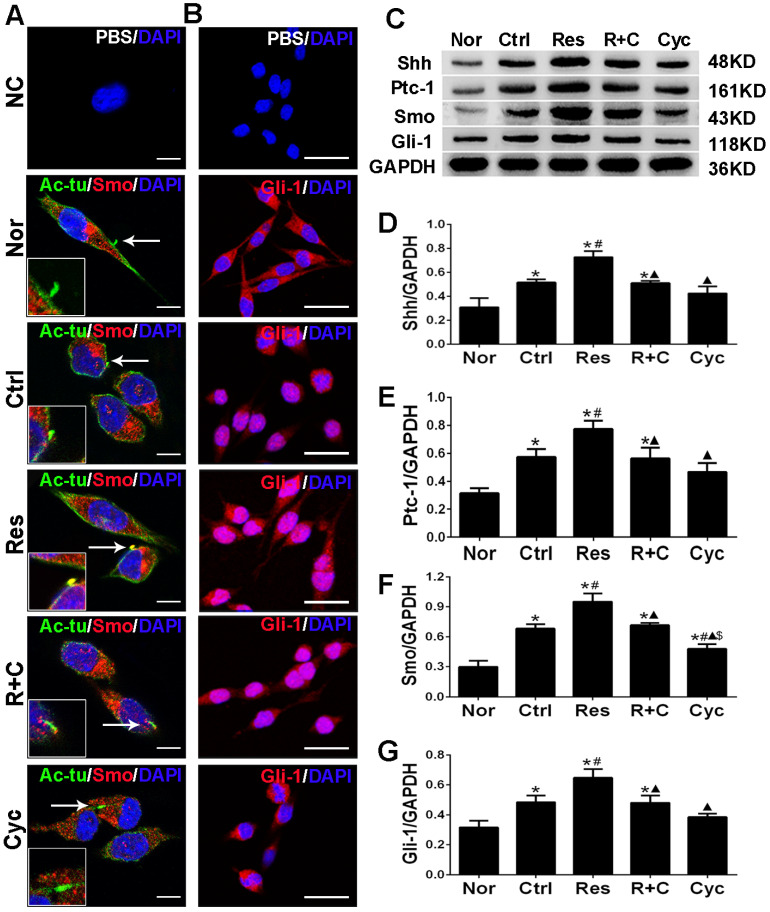
Cyclopamine reversed the effect of resveratrol on translocation of Smo/Gli-1 molecules and the Shh signaling in microglia after OGD/R injury in vitro. (**A**) Cyclopamine reversed the effects of resveratrol promoting Smo to translocate into primary cilia from cytoplasm after OGD/R injury in vitro. In the Nor, Ctrl, R+C and Cyc groups, Smo lay in cytoplasm. In the Res group, Smo translocated into the primary cilia. Scale bars = 10 μm. (**B**) Cyclopamine reversed resveratrol-promoting nuclear translocation of Gli-1 in microglia after OGD/R injury. In the Nor group, Gli-1 accumulated in cytoplasm. In the Ctrl group, Gli-1 partly transferred to the nucleus. In the Res group, Gli-1 almost transferred to the nucleus. However, cyclopamine combined with resveratrol (R+C) or cyclopamine alone (Cyc) inhibited the nuclear translocation of Gli-1. Scale bars = 50 μm. (**C**–**G**) Cyclopamine downregulated the expression of Shh, Ptc-1, Smo, and Gli-1 in microglia after OGD/R injury with Western blotting assay. * *p* < 0.05 vs. Nor; # *p* < 0.05 vs. Ctrl; ▲ *p* < 0.05 vs. Res; $ *p* < 0.05 vs. R+C (ANOVA, *n* = 3 each group).

## Data Availability

Not applicable.

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
