# Peer review of "Resveratrol Inhibits Activation of Microglia after Stroke through Triggering Translocation of Smo to Primary Cilia"

_jpm, 2023, doi:10.3390/jpm13020268_

Round 1

Reviewer 1 Report

The authors mechanistically investigate the resveratrol inhibitory effect on effect on activated microglia after stroke in vitro and in vivo

This work may be of readers' great interest, although many issues need to be addressed in this reviewer's opinion.

Authors only empathized Resveratrol's beneficial effects, while also adverse effects have been reported based on the dosage and the redox environment. For the sake of complete scientific information, both resveratrol-associated aspects should be reported. The reported paper may be of help (PMID: 32197410).

In this regard, how did the authors choose the used Resveratrol concentration?? What about higher or lower concentrations? The author should be aware that, mechanistically, Resveratrol's final outcome is tightly related to dosage and redox environment conditions. The reported papers may be of help in this regard (PMID: 31151226, PMID: 25656643, PMID: 23730364

In the provided immunofluorescence figures, how can the readers be sure that they are looking at the stated cells (e.g., microglia)? Each picture should have specific markers for the stated cells to ensure we are looking at these specific cell-related changes. Quantitive data of the reported immunofluorescence figures should also be provided to make possible statistical analysis and comparison.

The manuscript discussion should be completed with other resveratrol-associated mechanistic explanations that can provide neuroprotection. The papers below may be helpful in the discussion (PMID: 36237161, PMID: 35912113)

Reviewer 2 Report

The authors have presented an interesting paper on the role of resveratrol in the activation of microglia in the setting of cerebral infarction. In this work, they demonstrate that both in vivo with carotid occlusion and in vitro with cells subjected to oxygen and glucose deprivation, there is a significant reduction in microglia activation through different types of analyses. The results are interesting and support the proposed hypothesis.

Main concern:

Resveratrol has been proposed as a sirtuin activator and as a positive regulator of PPRgamma. The authors should provide a theory (experimentally based if possible) on how resveratrol activates Smo.

Minor concers:

Abstract: ‘…resveratrol inhibited activation…’ change by ‘…resveratrol partially inhibited activation…’

Abstract: ‘…target of resveratrol for inhibiting microglial activation…’ change by ‘…target of resveratrol for contributing to inhibit microglial activation…’

Introduction: ‘We definitively found that resveratrol inhibited activation and inflammation of microglia…’ change the whole sentence because it is the same that you have included at the end of the abstract and specify that the inhibition is partial.

Reviewer 3 Report

In the submitted manuscript (ID: jpm-2146388), the authors analysed the effect of food-bioactive compound resveratrol on microglia activation using the model system resembling the post-acute phase of stroke. They convincingly showed that resveratrol could inhibit the activation of microglia and inflammation and improve functional neurological outcomes through the modulation of the Hedgehog signalling pathway, triggering translocation of Smo to primary cilia. On the contrary, Smo antagonist cyclopamine cancelled these effects.

Essentially, it is a quality publication, well designed, realised, presented (results) and written (introduction, results and discussion). However, in its current state, the manuscript contains numerous technical flaws that spoil an otherwise very good overall impression. I ask the authors to seriously deal with it, considering that some of them have led to material mistakes!

1. The title: please change inhibit to inhibits! I suggest the authors review the text again and correct grammatical errors.

2. Latin words and expressions (via, in vitro, in vivo) are always better written in italics, starting from the abstract (lines 18, 20, 21, 23, 259) and continuing.

3. In the abstract and introduction section, the acronyms OGD/R and MCAO/R (injury types) are mentioned, but their meaning is stated only in the experimental part of the paper: please correct this.

4. Dozens of typesetting errors, so many that it would be difficult to list them all here. They are of the following types: lack of space between words (as in lines 40, 209, 216), then space between the number and the gossiping quantity, as in lines 86, 117, 119, 120, 127, 130, etc.). This led to chemically incorrect symbols for gases (for example, on line 107 and not only here in the text)! There are examples of extra spaces (two between words, as on lines 179, 295, 306, etc.) and non-uniform marking (space or not between the number and the percentage sign, for example, line 86 vs line 152). All in all, the entire text must be careful "technically" adjusted!

5. The division of experimental animals into groups is not the clearest (section 2.5.), probably also due to the reasons described in the point above. From what is written, there is no apparent difference between the sham and control group. Therefore, I suggest the authors schematically present the groups and how they were treated and replace the word drug (singular!) in the title of this chapter (resveratrol is considered a dietary supplement!). The origin of cyclopamine (manufacturer) is not specified in this section.

6. The listing of references 7, 14 (journal name needs to be added!!!), 26 and 31 need to be completed. Please correct it!

Round 2

Reviewer 1 Report

Unfortunately, the authors failed to discuss these critical points, which are at the basis of the final resveratrol outcome.

Authors only empathized Resveratrol's beneficial effects, while also adverse effects have been reported based on the dosage and the redox environment. For the sake of complete scientific information, both resveratrol-associated aspects should be reported as they are part of resveratrol behavior.

In this regard, how did the authors choose the used Resveratrol concentration?? What about higher or lower concentrations? The author should be aware that, mechanistically, Resveratrol’s final outcome is tightly related to dosage and redox environment conditions. Resveratrol-associated dose-response behavior should be discussed in the context of the doses used and the results obtained.

Reviewer 2 Report

The manuscript has improved and it can be accepted for publication

Author Response

Dear Reviewer,

    Thanks very much for your recommendation that our manuscript could be reconsidered for publication.

Reviewer 3 Report

The authors corrected the text according to the reviewers' suggestions in the revised version of the manuscript (jpm-2146388).

Therefore, I am writing to suggest that the current paper be accepted for publication; congratulations to the authors.

Author Response

(The authors gave the same response as above.)
